# Comment on "Opinion: Can uncertainty in climate sensitivity be narrowed further?" by Sherwood and Forest (2024)

**Nicholas Lewis**

independent researcher: Bath, UK

**Correspondence:** Nicholas Lewis (nhlewis@btinternet.com)

**Abstract.** This comment addresses assertions made by Sherwood and Forest (2024) (SF24) regarding the narrowing of the range of equilibrium climate sensitivity (ECS). SF24 challenged a previous study by Lewis (2022) (L22) that found a narrower and substantially lower ECS level. This comment clarifies that, contrary to SF24's claims, L22 did not rule out a high ECS level based on historical evidence and did identify and correct errors in Sherwood et al. (2020), in particular in relation to its likelihood estimation; their method, ironically, substantially underestimated likelihood for their historical evidence at high ECS levels. It also appraises L22's revisions to S20's methods and input assumptions and considers how these have contributed to the lowering and narrowing of the ECS range. This comment also discusses the role of priors in Bayesian ECS estimation and explains why the subjective Bayesian approach favoured by SF24 can often produce unreliable inference for uncertain parameters such as ECS. Finally, the importance of considering structural uncertainties in climate models, particularly regarding tropical warming patterns, is extended beyond the points raised by SF24. Such uncertainties could affect ECS estimation, not only from historical period evidence, but also from climate process understanding and emergent constraints. They seem more likely to suggest that existing ECS estimates are too high rather than too low.

## 1 Introduction

In the Sherwood and Forest (2024) opinion article entitled "Can uncertainty in climate sensitivity be narrowed further?" published in *Atmospheric Chemistry and Physics* (hereafter SF24), the authors express doubts that the uncertainty range for equilibrium climate sensitivity (ECS) has been further narrowed since the publication of Sherwood et al. (2020) (hereafter S20). They note that the observationally driven ECS range in S20 was approximately adopted in the relevant chapter (Forster et al., 2021) of the IPCC Sixth Assessment Report (AR6).

SF24's authors state that the new study claiming the largest revision in the range for ECS is Lewis (2022) (hereafter L22), which "asserts a narrower and substantially lower ECS level using the basic S20 methodology with various updates". This comment addresses the erroneous claims that SF24 made about L22. It identifies L22's contributions – through revisions to S20's input assumptions for each line of evidence and other changes – to the lowering and narrowing of S20's ECS range. It addresses problems with the subjective Bayesian approach described by SF24 and finally discusses the challenges in narrowing the range of climate sensitivity posed by uncertainty as to the realism of global climate model (GCM)-simulated long-term tropical warming patterns. For clarity, it should be noted that while SF24 refers to ECS, S20 and L22 actually both estimate $S$, a proxy for ECS. In GCMs, $S$ is almost always slightly lower than ECS. The two terms are only distinguished herein when discussing their relationship.

## 2 Critique of SF24's claims regarding L22

SF24 states the following concerning L22.

> While this author claims "errors" in S20, looking carefully it appears these are differences in opinion on methodological choices and priors rather than errors, and they moreover were acknowledged to have little effect on the outcome.

There were indeed differences in opinion on Bayesian priors between L22 and S20. And S20's decision to not adjust for $CO_2$ forcing increasing slightly faster than logarithmically with concentration, unlike L22, is arguably a defensible difference of opinion, although its application led to inconsistency between S20's estimates from different lines of evidence. The minor effects on ECS estimation of these two differences of opinion almost cancelled out.

### 2.1 L22's findings of errors, inconsistencies, and indefensible methodological choices in S20

However, L22 also found actual errors and indefensible methodological choices in S20.

L22 Sect. 5.1 and their Supplement (Sect. S2) showed that S20's method of likelihood estimation was invalid and resulted in a major underestimation of the historical evidence likelihoods at high climate sensitivities. S20's likelihood estimates based on Paleocene–Eocene Thermal Maximum (PETM) evidence were similarly affected, although PETM evidence was not used in the main S20 results. Moreover, L22 pointed out that S20 used an uncertainty estimate that was a factor of 10 lower than stated for PETM $CO_2$ forcing due to a coding error[1].

Moreover, L22 Sect. 4.1 and their Supplement (Sect. S1) showed that S20's ECS estimates from both process evidence and historical evidence were biased high because, in deriving ECS from the climate feedback estimate, the authors had inappropriately used an estimate of the effective radiative forcing (ERF) for doubled $CO_2$ ($F_{2\times CO_2}$) based on land-warming-corrected fixed sea surface temperature (SST) simulations, the method subsequently used in AR6, rather than a 150-year-regression-based estimate. As L22 showed, GCM $F_{2\times CO_2}$ estimates based on the fixed SST simulation method (which for L22's ensemble of 26 GCMs agreed very closely, at their median, with the AR6 best estimate of $F_{2\times CO_2}$) are in general significantly higher than regression-based $F_{2\times CO_2}$ estimates. Regression-based $F_{2\times CO_2}$ estimates represent the $y$ intercept from a linear regression of planetary radiative imbalance ($N$) against the increase in global mean surface temperature ($\Delta$GMST) as simulated in a GCM over 150 years following an abrupt quadrupling in $CO_2$ concentration (abrupt4xCO2), scaled to a doubling of $CO_2$. The

$x$ intercept of the regression line – the definition of $S$ – is taken as the GCM's estimated ECS, and the regression slope is taken as an estimate of its net climate feedback. In almost all GCMs, climate feedback weakens over the first few decades of the 150-year simulation, reflecting evolving SST warming patterns (a forced pattern effect). That being so, $N$ is not linearly related to $\Delta$GMST, and such regression-based $F_{2\times CO_2}$ values are bound to underestimate true $F_{2\times CO_2}$, as estimated from fixed SST simulations[2]. However, estimates of $S$ (which is what S20 and L22 estimate in place of ECS) calculated by dividing observationally derived estimates of what the slope in a 150-year abrupt4xCO2 simulation regression would be into an estimate of $F_{2\times CO_2}$ require, to be arithmetically correct, that the $F_{2\times CO_2}$ estimate used is likewise a regression-based one. Therefore, S20's use of a fixed-SST-based, rather than a regression-based, $F_{2\times CO_2}$ estimate will have biased-high its process- and historical-evidence-derived estimates of $S$, which were based on such calculations. Based on the median ratio of those two types of $F_{2\times CO_2}$ estimates in the GCM ensemble, L22 estimated the high bias to be approximately 16 %.

### 2.2 SF24's criticisms of L22

SF24 criticises L22's results by claiming the following.

> Instead, the reduction and narrowing of the ECS probability density function (PDF) resulted from a selective use of evidence – most importantly, a decision to reject the possibility of a large "pattern effect" on historical sea surface temperature (SST), even though this continues to be strongly supported by new studies, and a downward revision of expected historical aerosol cooling. Together these two departures allowed Lewis to conclude (in contrast to other studies) that the historical record rules out a high ECS level.

This claim is totally wrong. On the contrary, L22 finds (Table 8) that the historical record does not rule out a high ECS level. The standard 90 % (5 %–95 %) uncertainty range that L22 arrived at using only data from the historical record was 1.2–7.6 °C. Where, in addition, the common prior assumption that ECS is positive and does not exceed 20 °C was made, the range became 1.15–6.1 °C. Neither of these ranges rule out a high ECS level. Both 7.6 and 6.1 °C substantially exceed the 4.7 °C 95 % uncertainty bound of S20's main combined-evidence ECS estimate.

S20's combined-evidence median (50 % probability) baseline ECS estimate was 3.1 °C, with 66 % ("likely") and

---

[1]Since the publication of L22 this coding error has been noted, and its effects corrected, in the online version of S20.

[2]Reflecting the cause of this underestimation, the data in L22 Table S1 in their Supplement show a 0.98 TS1 correlation across the 26 GCMs between the measure of the forced pattern effect and the degree to which regression-based $F_{2\times CO_2}$ underestimates fixed-SST-based $F_{2\times CO_2}$.

90 % ranges of respectively 2.6–3.9 and 2.3–4.7 °C. The corresponding values in L22 were 2.16, 1.75–2.7, and 1.55–3.2 °C. Without the two revisions criticised by SF24, the 5 % bound and median for the L22 combined-evidence ECS estimates would have changed by less than 0.05 °C and the 95 % bound would have increased by only about 0.1 °C. Excluding historical evidence entirely would similarly have left the 5 % bound and median estimates unchanged, with the 95 % bound increasing by only 0.2 °C. Thus the reduction and narrowing of the ECS PDF in L22 had almost nothing to do with the revisions it made to assumptions about the pattern effect or aerosol cooling over the historical period.

## 2.3 Historical aerosol forcing estimates and uncertainty

In both S20 and L22, aerosol forcing is a major source of uncertainty for historical evidence. It is therefore unsurprising that the historical record does not rule out a high ECS level. Aerosol forcing reduces, to an uncertain extent, the magnitude of the denominator of the energy budget formula for estimating ECS. That results in there being a non-negligible probability, indeed a substantial one based on S20's assumptions, that the denominator is small or negative, implying a very high or unbounded ECS but a negligible probability that the denominator is very large, thus ruling out ECS being very low. It is surprising that the SF24 authors failed to recognise that it is process and, particularly for S20, paleoclimate evidence that most constrained the ECS upper bound both in L22 (Table 8) and in S20 (Table 10).

Both S20 and L22 also estimated an effective climate sensitivity over the historical period ($S_{hist}$), which is not adjusted for any pattern effect and is somewhat less affected than ECS by assumptions about aerosol cooling. Employing a sampling method, S20 derived a 90 % range for $S_{hist}$ of 1.9–14.4 °C, with a median of 3.1 °C, using an aerosol forcing distribution that assigned a 16 % probability to it being even stronger (more negative) than $-2.0\,\mathrm{W\,m^{-2}}$. L22 down-weighted the probability of very strong aerosol forcing from that assumed in S20, but without assuming that the most likely level of aerosol forcing was weaker than in S20, and derived a 90 % range for $S_{hist}$ of 1.3–4.3 °C (median 2.1 °C). As noted in S20, without criticism, Tokarska et al. (2020) likewise effectively down-weighted very strong aerosol forcing as less consistent with observations. That resulted in their 90 % range for $S_{hist}$ being 1.3–3.1 °C, with a median of 2.1 °C – identical results to L22 except that the 95 % bound in L22 was considerably higher.

## 2.4 Historical pattern effect estimates

The pattern effect – the dependence of outgoing radiation to space on the geographical pattern of SST warming – is generally thought to result in ECS being higher than $S_{hist}$, although some studies implicitly question whether this is so (see discussion in Sect. 5). The reduction in the tropical Pacific east–west temperature gradient that occurs after a decade or two in almost all GCM $CO_2$-forced warming simulations, with greater warming occurring in the east than in the west, underlies the weakening over time of net climate feedback in GCMs (the forced pattern effect), which is what causes their ECS values to be higher than implied by their response over multidecadal periods. Moreover, during the historical period internal variability is generally thought to have caused an additional, unforced pattern effect, particularly over the last few decades. Of the three studies that SF24 cites as supporting a large total historical pattern effect, neither Heede and Fedorov (2021) nor Chao et al. (2022) consider the full historical period (from the second half of the 19th century on), which is what S20 and L22 both use. They only consider post-1980 and post-2000 periods, respectively, and thus do not in fact support SF24's claims. The third study cited by SF24, Andrews et al. (2022a), did estimate the pattern effect over the full historical period (1871–2010), regressing annual mean historical simulation data from atmosphere-only GCMs driven by SST evolution. Their result when using data from the non-spliced HadiSST1 dataset rather than the outlier spliced AMIPII SST dataset, of $0.48\,\mathrm{W\,m^{-2}\,°C^{-1}}$, is in line with the S20 estimate of $0.5\,\mathrm{W\,m^{-2}\,°C^{-1}}$. Their estimate is reduced to $0.41\,\mathrm{W\,m^{-2}\,°C^{-1}}$ when regressing pentadal mean data, an approach that suppresses bias from responses to interannual fluctuations (Lewis and Mauritsen, 2021). If the resulting individual model estimates are weighted equally by modelling centre rather than by model, recognising that models from the same centre have structural similarities, their estimate is reduced further to $0.36\,\mathrm{W\,m^{-2}\,°C^{-1}}$, in line with the L22 estimate of $0.35\,\mathrm{W\,m^{-2}\,°C^{-1}}$. Moreover, Modak and Mauritsen (2023) obtained an even lower historical pattern effect estimate of $0.30\,\mathrm{W\,m^{-2}\,°C^{-1}}$, averaged across seven SST datasets. They also noted that, of the SST datasets that they studied, the commonly used AMIPII dataset produced by far the largest pattern effect estimate.

## 3 Effects on ECS estimates of L22's various revisions to S20's assumptions

It is relevant to appraise the contributions made to the lowering and narrowing of the ECS range in L22 by the various revisions made in it to S20's methods and input assumptions for each of the four lines of evidence that they both used: process, historical, paleoclimate cold period, and paleoclimate warm period. The impact of correcting S20's likelihood estimation and of the change in prior is shown in Table 4 of L22 for all cases where S20 provides posterior PDFs, and the impact on measures of the posterior PDF of different input assumptions between L22 and S20 on ECS estimates from each line of evidence on its own are detailed in Table 8 of L22. The effects of key classes of data differences on L22's combined-evidence ECS estimation are set out in Table 7 of

L22; the effects of revisions to S20's input assumptions by line of evidence are set out in detail below.

The cumulative effects of these changes are summarised in Table 1. All percentage reductions in the median ECS estimate shown in Table 1 or mentioned below are from the 3.23 °C estimate using L22's Jeffreys prior and after correcting S20's likelihood errors. The changes made in L22 to S20's historical pattern effect and aerosol forcing magnitudes and to process cloud feedback are more open to debate than are the other input data revisions, which should not be regarded as particularly contentious, and their effects are therefore shown separately. Even without revisions in S20's estimates of any of those three input items, the bulk of the reduction from S20 to L22 in the ECS estimate range and over 80 % of the reduction in the median estimate are still obtained.

### 3.1 Effect of adopting a computed Jeffreys "objective" prior and correcting S20's likelihood estimates

As shown in the top section of Table 1, L22's replacement of S20's prior distribution with a computed Jeffreys prior, combined with the correction of S20's likelihood estimation method, widened and slightly raised the posterior ranges for ECS. Changing the prior and correcting the likelihoods made approximately equal contributions to the change in the ECS posterior median and 5 %–95 % ranges.

### 3.2 Effect of updating the $F_{2\times CO_2}$ estimate, adjusted to a regression basis where appropriate, and revising the ECS to $S$ ratio

L22 incorporated the ratio ($\gamma$) of regression-based to fixed-SST-based estimates of $F_{2\times CO_2}$ when using process or historical evidence (as is undoubtedly appropriate and necessary to avoid them being on a basis that is inconsistent with estimates of $S$ from paleoclimate evidence: Sect. 2.1 and L22 Sect. 4.1) and updated S20's 4.00 W m$^{-2}$ fixed-SST-based estimates of $F_{2\times CO_2}$ to the 3.93 W m$^{-2}$ AR6 value. L22 also used an ECS to $S$ $(1 + \zeta)$ ratio estimated by comparing their values in each of 16 long $CO_2$ doubling or $CO_2$ quadrupling simulations (L22 Supplement, Sect. 5.3.1), rather than using S20's ECS to $S$ ratio (which resulted in inconsistent estimation of $S$ between different lines of evidence in S20: L22 Sect. 2). These methodological choices reduce the median ECS estimate to 2.82 °C, accounting for 38 % of the overall reduction in ECS (to 2.16 °C) from all the L22 input assumption revisions combined.

### 3.3 Effect of AR6-based revisions to the historical surface air temperature to blended warming ratio and non-aerosol forcings

Substituting updated estimates from IPCC AR6 of the ratio of historical global near-surface air temperature (SAT) warming to blended SST and land SAT warming and of changes in historical forcings other than for aerosols, while retaining S20's aerosol forcing distribution, further reduces the estimated ECS range. The resulting cumulative reduction in the median ECS estimate to 2.64 °C represents 55 % of the overall L22 reduction. None of the foregoing data input revisions can reasonably be regarded as contentious.

### 3.4 Effect of revisions to cold paleoclimate evidence input assumptions

The revisions made in L22 to S20's estimates of cooling and non-greenhouse-gas forcing at the Last Glacial Maximum (LGM), discussed in detail in L22's Supplement (Sect. 5.3.2), are also strongly defensible. The downward revision to estimated LGM cooling still left it greater than the average of the estimates from S20's cited sources, adjusted to correctly reflect the preindustrial-to-LGM temperature change. Moreover, L22's revised cooling estimate is identical to the best estimate in Annan et al. (2022). That study followed two LGM cooling studies (Tierney et al., 2020, and Osman et al., 2021) that estimated stronger cooling. All three studies combined information from GCM simulations and similar sets of proxies, but only Annan et al. (2022) used a method that prevented the global mean magnitude of cooling (as opposed to the spatial pattern of cooling) in the GCM simulations from biasing the estimate of LGM cooling.

The revision to LGM non-greenhouse-gas forcing adds, to the land ice and sea level forcing, an estimate of the omitted albedo change caused by sea level fall exposing more land; S20 claimed this was a "less commonly discussed factor", but as L22 pointed out multiple studies have accounted for it.

Adopting these two changes results in a further 0.13 °C reduction in the median ECS estimate to 2.51 °C, a cumulative fall representing 67 % of the overall L22 reduction.

It is also relevant to note that while L22 retained S20's assumption that ECS was lower at the LGM than for warming from the preindustrial climate, and accordingly applied an upwards adjustment when estimating the latter, a recent study finds the opposite to be true (Cooper et al., 2024). Replacing S20's and L22's upwards adjustment to ECS with an adjustment for that study's finding would have significantly lowered the LGM-based climate sensitivity estimates in both S20 and L22.

### 3.5 Effect of revisions to warm paleoclimate evidence input assumptions

The revisions made in L22 to S20's estimates of the ratio of Earth system sensitivity (ESS) to ECS, and the ratio of global SAT warming to tropical SST warming, for the mid-Pliocene warm period (mPWP) (discussed in detail in L22's Supplement, Sect. 5.3.3) should likewise not be seen as contentious re-interpretations. S20 used mPWP simulation results from the PlioMIP1 model ensemble to estimate

**Table 1.** Evolution from the S20 to the L22 combined-evidence estimates of $S$ (the S20 and L22 proxy for ECS) arising from the change in prior and correction of likelihood estimation (top section), as well as subsequent changes arising from revising input data assumptions. All values are in degrees Celsius (°C) and except for the medians are rounded to the nearest 0.05 °C. Medians, as the key measure of changes in the posterior PDF for $S$, are shown in bold font.

| Percentile of posterior PDF for $S$ (as %) | 5 % | 50 % (median) | 95 % | Fall in median/total reduction |
|---|---|---|---|---|
| Baseline results in S20 | 2.3 | **3.1** | 4.7 | |
| Corresponding L22 estimates after correcting S20's likelihoods and adopting a computed Jeffreys prior | 2.3 | **3.23** | 5.05 | 0 % |
| *L22 estimates after then adopting revisions relating to the following:* | | | | |
| derivation of the ECS / S ratio $(1 + \gamma)$, adjustment of $F_{2 \times CO_2}$ to a regression basis $(\zeta)$, and adopting the AR6 $F_{2 \times CO_2}$ estimate | 1.95 | **2.82** | 4.35 | 38 % |
| also AR6 historical ex-aerosol ERF and $\Delta T$ basis[a] | 1.85 | **2.64** | 4.1 | 55 % |
| also LGM $\Delta T$ and $\Delta F_{exCO_2}$ (preferred evaluation) | 1.7 | **2.48** | 3.9 | 67 % |
| also mPWP GMST/tropical SST and ESS / ECS ratios | 1.65 | **2.36** | 3.65 | 81 % |
| also process Planck feedback | 1.65 | **2.34** | 3.6 | 83 % |
| also process cloud feedback | 1.55 | **2.19** | 3.3 | 97 % |
| also historical pattern effect | 1.55 | **2.15** | 3.25 | 101 % |
| also historical aerosol ERF (all revisions) | 1.55 | **2.16** | 3.2 | 100 % |

[a] AR6 historical ERF time series are used to estimate $\Delta F_{\text{ex-aerosol}}$, but only to scale the main 1850 to 2005–2015 $\Delta F_{\text{aerosol}}$ estimate to a 1861–1880 to 2006–2018 change. The AR6 revision to the $\Delta T$ basis relates to changing S20's estimated GMAT–GMST adjustment to match the AR6 zero estimate of their difference.

the ESS-to-ECS ratio. The revised L22 estimate used results from the more recent PlioMIP2 model ensemble instead. L22 also used PlioMIP2 mPWP simulations to estimate the global SAT to tropical SST warming ratio, whereas S20 used pre-PlioMIP2 studies and simulations that estimated the ratio for glacial cycles, which involve climate states very different from that in the mPWP. L22 also increased the width of S20's mPWP global SAT uncertainty range, in view of proxy evidence being sparse. Adopting these changes results in a further 0.15 °C reduction in the median combined-evidence ECS estimate to 2.36 °C, thus achieving 81 % of the overall L22 reduction. The 5 %–95 % range becomes 1.65–3.65 °C. By this point 76 % of the total reduction in the 95 % uncertainty bound has been achieved.

### 3.6 Effects of revisions to process evidence input assumptions

The revisions made in L22 to S20's Planck and cloud feedback estimates further reduce the L22 combined-evidence ECS estimate, with the median falling to 2.19 °C, thus achieving 97 % of L22's overall reduction. The revision to S20's median Planck feedback estimate was justified in L22's Supplement (Sect. 5.1.2) and was very small: from $-3.20$ to $-3.25 \, \text{W m}^{-2} \, °\text{C}^{-1}$. By comparison, the average of the estimates for CMIP5 and CMIP6 models in Zelinka et al. (2020) was $-3.28 \, \text{W m}^{-2} \, °\text{C}^{-1}$.

Adopting L22's minor revision to Planck feedback but not its larger revision to S20's median cloud feedback estimate reduces the median ECS estimate only very marginally to 2.34 °C (thus realising 83 % of L22's total reduction in the median estimate), with the 5 %–95 % range becoming 1.65–

3.6 °C. The revision in L22 to S20's cloud feedback estimate (from 0.45 to 0.27 $\text{W m}^{-2} \, °\text{C}^{-1}$) and its effects are discussed in Sect. 3.8.

### 3.7 Effect of revisions to the historical aerosol forcing and pattern effect input assumptions

After adopting L22's revisions L22 to S20's cloud and Planck process feedbacks, the only remaining revisions made in L22 were to S20's historical aerosol forcing and pattern effect estimates. The adoption of these final two revisions has a remarkably small impact on L22's combined-evidence ECS estimate. The reduction to the original L22 median estimate is only 0.03 °C, while the 5 % uncertainty bound is unchanged and the 95 % bound is reduced by 0.1 °C. Even after L22's revisions, historical evidence remained a weak constraint on high ECS because, as L22 Fig. 5 shows, at L22's combined-evidence 95 % bound of 3.2 K its historical likelihood had not declined very far from its peak value, and even at higher climate sensitivity values the rate of its decline was very much slower than for the likelihood from either process evidence or paleoclimate evidence. The reason why L22's revisions to S20's historical aerosol forcing and pattern effect estimates had little effect on its median combined-evidence ECS estimate is that before making them the historical likelihood was not, unlike L22's process and paleoclimate likelihoods, changing sharply in the region of the combined-evidence likelihood maximum, so it had little effect on the combined-evidence median ECS estimate, and after making those changes the historical likelihood maximum was in line with L22's process and paleoclimate likelihood maxima, so it also had little effect on the median ECS estimate.

### 3.8  Effects of various combinations of revisions to the most uncertain input assumptions

Cloud feedback remains quite poorly constrained and is the dominant source of process evidence feedback uncertainty. While SF24 does not explicitly challenge L22's cloud feedback estimate, which was smaller than in S20 due to L22 adopting a lower estimate of low-cloud feedback, it does cite a subsequent study (Stauffer and Wing, 2022) that suggests that non-low cloud feedback – the S20 estimate of which was not revised in L22 – is higher than assumed in S20 (and hence in L22). It also cites a post-S20 study (Ceppi and Nowack, 2021) that estimates almost the same total cloud feedback as in S20, although with an even smaller low-cloud feedback than assumed in L22. The revision of the cloud feedback estimate in L22 should therefore be regarded as subject to significant uncertainty. Moreover, historical aerosol forcing is poorly constrained, as discussed in Sect. 2.3, and L22's revision of it may also be regarded as subject to significant uncertainty. As discussed in Sect. 2.4 the revision in L22 of S20's historical pattern effect appears to be well justified, but in view of SF24's focus on this point it is treated here as being subject to significant uncertainty.

If no changes are made to S20's cloud feedback or historical aerosol forcing estimates, but all the other revisions in L22 are retained, the effect on L22's combined-evidence ECS estimate is fairly modest: the original L22 median estimate increases by about 0.15 to 2.32 °C, while the 5 %–95 % range becomes 1.6–3.55 °C, an increase of 0.35 °C at the top end. These ECS estimate increases barely change if S20's historical pattern effect estimate is left unaltered as well as S20's cloud feedback and historical aerosol forcing estimates (see Sect. 3.6).

If no changes are made to S20's cloud feedback or historical pattern effect estimates, but all the other revisions in L22 are retained, the original L22 median estimate increases by about 0.2 to 2.37 °C, while the 5 %–95 % range becomes 1.65–3.65 °C.

If only cloud feedback is reverted to S20's estimate, retaining all the other L22 revisions, the effects are much the same, with the median L22 ECS estimate increasing to 2.28 °C and the 5 %–95 % range becoming 1.6–3.45 °C.

If only the historical pattern effect is reverted to S20's estimate, retaining all the other L22 revisions, the effects are smaller, with the median L22 ECS estimate increasing to 2.21 °C and the 5 %–95 % range becoming 1.55–3.3 °C.

If instead only aerosol forcing is reverted to S20's estimate, the original L22 ECS estimates are essentially unchanged, with just the 95 % uncertainty bound marginally increasing from 3.2 to 3.25 °C. As explained in L22 Sect. 6, the minor impact of reverting to S20's aerosol forcing estimate is due to the main effect on the historical likelihood of doing so being to increase it at higher ECS values, where the combined-evidence likelihood is strongly down-weighted by low process and paleoclimate likelihoods.

## 4  Subjective vs. objective Bayesian approaches

Regarding statistical issues, as noted by SF24 priors on ECS and other climate system parameters have been a contentious issue since the first Bayesian ECS studies. SF24 argues that probability is most useful as a quantification of what someone expects, rather than a quasi-objective calculation based on the chosen physical models and data used. In other words, they favour a subjective Bayesian rather than an objective Bayesian or a frequentist approach. However, as explained in L22, it is essential for scientific inference that the statistical methods used are calibrated, in the sense that the uncertainty ranges they generate closely approximate confidence intervals. Objective Bayesian methods involve use of mathematical, noninformative priors that are generally intended to produce uncertainty ranges that are (at least approximately) true confidence intervals; frequentist methods share that intention. Subjective Bayesian methods are not designed to do so, and their uncertainty ranges may be very ill-calibrated when (as in ECS estimation) the data are insufficiently strong to dominate the influence of the prior. S20 used a subjective Bayesian approach, while L22 employed an objective Bayesian one using a computed Jeffreys prior. Doing so widened and slightly raised the combined-evidence posterior ranges for climate sensitivity, before making any changes to S20's other assumptions.

There are also other problems with adopting a subjective Bayesian approach. First, the prior and data likelihood may well both be based on the same evidence, at least to some extent. Scientists' expectations as to the value of ECS can only rationally be based on some set of observational evidence and/or on climate model behaviour. It would be reasonable to use the well-established fact that the climate system is not extremely unstable to require a prior to rule out ECS being negative or exceedingly high (over 20 °C, say). However, a considerable part of the observational data-based evidence available to scientists will already be reflected in the likelihood function(s) produced by the physical models and data used for estimating ECS, so it is duplication to also use it in formulating a prior. Moreover, since observational evidence is used in the construction of climate models, and because related aspects of climate model behaviour will have been used to quantify the data-variable distributions used in estimating the ECS likelihood, climate model behaviour should not shape the prior, particularly given the concern that ECS in such models may be unrealistic.

Secondly, even if the chosen prior does produce well-calibrated uncertainty ranges when used to infer ECS from a data likelihood reflecting one set of evidence, the standard Bayesian method of simply updating that initial posterior PDF by data likelihoods reflecting other sets of evidence may well not produce well-calibrated uncertainty ranges reflecting the combined evidence (Lewis, 2018; Lewis and Grunwald, 2018).

For historical and, particularly, process evidence, the uniform (flat) prior for climate feedback that S20 selected was in fact close to the mathematical, noninformative prior used in L22, although for paleoclimate evidence it differed substantially (L22 Fig. 3). However, the uniform prior for ECS selected for use with observational evidence in the IPCC Fourth Assessment Report was very far from noninformative and resulted in huge overestimation of the chance of ECS being very high; only the upper bound placed on that prior (20 °C for most studies) prevented almost 100 % probability from being assigned to near-infinite ECS values. The same applies to S20's ECS estimate from historical evidence when using a 0–20 °C uniform prior for ECS (S20 Table 6); the median estimate was 8.5 °C, with a 95 % bound (existing purely due to the prior upper limit of 20 °C) of 18.6 °C. By contrast, when using a noninformative prior with S20's historical evidence inputs, the L22 (Table 4) median ECS estimate was only 4.2 °C, with a 95 % bound of 13.7 °C, despite incorporating correction of S20's underestimation of its historical evidence likelihood at high ECS levels.

## 5  Structural uncertainties in climate models

SF24 raises the issue of structural uncertainty in GCMs and other forward models and notes the inability of GCMs to simulate the historical pattern of warming, including changes in the Pacific east–west temperature gradient. This failure could be because the pattern of Pacific warming is a transient phenomenon or a result of missing aerosol forcing mechanisms. However, other possible explanations could have very different ramifications for ECS. Several recent studies suggest that GCM-simulated weakening of the Pacific east–west temperature gradient may be unrealistic, with SST in the West Pacific Warm Pool being more sensitive to greenhouse gas forcing than SST in the eastern equatorial Pacific (Seager et al., 2019; Lee et al., 2022; Hou et al., 2024; Lee et al., 2024). If correct, the weakening of net climate feedback in GCMs over 150 years after an abrupt quadrupling of $CO_2$ concentration would also be unrealistic. That would imply both that current estimates of the historical pattern effect (Sect. 2.4) are excessive and that most current ECS estimates, including those based on process understanding and on emergent constraints as well as on historical warming – which (like those in S20 and L22) generally reflect the aforesaid weakening of climate feedback in GCMs, may be significantly biased upwards.

The possibility that long-term tropical warming patterns simulated by GCMs are significantly wrong could be one of the most important of the omitted structural uncertainties in ECS estimation about which SF24 expresses concern. However, that uncertainty points primarily to the possibility that ECS estimates are too high, not too low. Even if the Pacific east–west temperature gradient, and hence net climate feedback, does eventually weaken to the extent simulated by GCMs, a multidecadal to centennial delay in that weakening occurring could imply a significantly lower warming response this century, as a fraction of ECS, than would otherwise be the case.

SF24 claims that probability distributions for ECS in S20 and AR6 remain approximately valid but that subsequent studies, including L22, omit important structural uncertainties. It is possible that both S20 and L22 omitted such uncertainties, but L22 did not omit any structural uncertainties that were included in S20. The equations used in L22 for estimating ECS were identical to those in S20, except for the inclusion of an additional structural uncertainty concerning the ratio of regression-based to fixed-SST-based estimates of $F_{2 \times CO_2}$. Moreover, the uncertainty estimates for each input variable in L22 were almost all the same as or greater than those used in S20, the main exception being for historical aerosol forcing, the changed input distribution of which had a negligible effect on the L22 combined-evidence ECS estimation.

## 6  Conclusions

S20 substantially narrowed the uncertainty in ECS, primarily by rejecting lower values of ECS that had been included in climate assessments since 1979 and retained through to the 2013 IPCC Fifth Assessment Report. Its observationally driven ECS approach and range was approximately adopted in IPCC AR6. By remedying deficiencies in the S20 analysis and adopting input assumptions based on newer evidence, and in a few cases alternative appraisals of existing evidence, L22 provided an analysis that re-emphasised the lower values of ECS. While S20 estimated the probability of ECS being below 2.3 °C as only 5 %, L22 estimated it to be over 50 %. S24's criticisms of assertions in L22 regarding S20 and of L22's ECS estimation are unjustified and incorrect.

The reluctance of climate scientists to move away from mathematically unsound subjective Bayesian approaches for estimating ECS or other uncertain climate system parameters is deeply concerning. Ensuring reliable results requires either adopting an objective Bayesian approach, employing a noninformative prior, or using frequentist statistical methods.

While structural uncertainty in forward models, in particular GCMs, may well be underestimated, SF24's claim that probability distributions for ECS in S20 remain approximately valid but that subsequent studies omit important structural uncertainties cannot be justified with respect to L22.

**Code and data availability.** Simulation data used to re-estimate HadiSST1-driven pattern effect results in Andrews et al. (2022a) using pentadal mean ordinary least-squares regression were downloaded from https://doi.org/10.5281/zenodo.6799004 (Andrews et al., 2022b) on 15 February 2025. The regression code used was the lm function in R (R Core Team, 2025).

**Competing interests.** The author has declared that there are no competing interests.

**Disclaimer.** Publisher's note: Copernicus Publications remains neutral with regard to jurisdictional claims made in the text, published maps, institutional affiliations, or any other geographical representation in this paper. While Copernicus Publications makes every effort to include appropriate place names, the final responsibility lies with the authors.

**Acknowledgements.** The author acknowledges Timothy Andrews for publicly archiving global annual mean simulation data used in Andrews et al. (2022a).

**Review statement.** This paper was edited by Ken Carslaw and reviewed by one anonymous referee.

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

## Remarks from the typesetter

TS1    Changes in values (0.98 to 0.9) require editor approval. If you would like to change the number, please provide a short explanation regarding the correction that can be forwarded by us to the editor.

TS2    Not mentioned in the text (Rugenstein et al., 2020).