# Peer review of "Comment on "Can uncertainty in climate sensitivity be narrowed further?" by Sherwood and Forest (2024)"

_EGUsphere, 2025_

## Author Response (AR1)

**Final response to the Editor re manuscript egusphere-2025-1179**

Text in the comments that requires a response from the author is shown in italics. In addition to revisions made in response to comments RC1 and CC1, the changes include a large number of minor edits and other rewordings designed to make the manuscript clearer and easier to understand.

Nicholas Lewis

**Responses to referee comment RC1**

I thank the anonymous referee for his helpful review of my submitted manuscript (L25). My responses to points in it requiring input from the author (which I have numbered for ease of reference) are as follows.

1. *LF24 cited some evidence in favour of the pattern effect. L25 acknowledges it but draws on other studies not cited by LF24 which provide counterevidence, and I think readers will find the exchange illuminating.*

I thank the referee for this comment, in the light of which I have now bolstered the discussion of the pattern effect by adding the following sentence at the end of the relevant subsection (2.4), citing another post L22 study that provides counterevidence to SF24's arguments in favour of a large historical pattern effect:

"Moreover, Modak and Mauritsen (2023) obtained an even lower historical pattern-effect estimate of 0.30 $Wm^{-2}{}^{\circ}C^{-1}$, averaged across seven SST datasets. They also noted that, of the SST datasets that they studied, the commonly used AMIPII dataset produced by far the largest pattern effect estimate."

2. *However the discussion in L25 is broken up, occurring in lines 90-100 and then in lines 139-157, so editing is needed to combine these into a single discussion.*

I have carefully considered combining the discussion in lines 90-100 and in lines 139-157, but I have concluded that it would be unhelpful to the reader to fully merge these two discussions. The discussion in lines 90-100 specifically concerns whether, in the light of SF24's criticism of it, the reduction in L22 of the estimated historical pattern effect – which includes both a forced and an unforced (internal variability caused) pattern effect – was justified by evidence in the historical pattern effect literature, including in particular the studies cited in SF24. The discussion in lines 139-157 concerns the forced pattern effect in abrupt4xCO2 simulations, in the context of a discussion about structural uncertainties in GCMs. This involves different theories of the development of the tropical Pacific east-west SST gradient under $CO_2$ forcing, which strongly affects the forced pattern effect (but not the unforced pattern effect) and could even indicate that it doesn't exist. Including this discussion within that in lines 90-100 (now in section 2.4) would make that discussion very lengthy and less focused on refuting SF24's criticisms of L22, which is what section 2 concerns. Equally, the discussion of the pattern effect in section 2 cannot sensibly be left until lines 139-157 (now in section 5).

I have therefore instead: a) moved, with some changes, the third sentence of section 5, which introduces and defines the forced pattern effect and explains its origins, to section 2.4; b) added an extra sentence in section 2.4 about the unforced element of the historical pattern effect; and c) added a cross reference where the pattern effect is discussed in section 2.4 to the related (but different) discussion of the pattern effect in section 5, and vice versa. I have also rationalised and simplified the discussion in section 5 of studies questioning the forced pattern effect.

The affected text at the start of section 2.4 now reads:

"The pattern effect – the dependence of outgoing radiation to space on the geographical pattern of SST warming – is generally thought to result in ECS being higher than $S_{hist}$, although some studies implicitly question whether this is so (see discussion in Sect. 5). The reduction in the tropical Pacific east-west temperature gradient that occurs after a decade or two in almost all GCM $CO_2$-forced warming simulations, with greater warming occurring in the east than in the west, underlies the weakening over time of net

climate feedback in GCMs (the forced pattern effect), which is what causes their ECS values to be higher than implied by their response over multidecadal periods. Moreover, during the historical period internal variability is generally thought to have caused an additional, unforced, pattern effect, particularly over the last few decades."

and in the first paragraph of section 5, in addition to its original third sentence having been removed and various changes made to rationalise and improve the wording, the final sentence now reads:

"That would imply both that current estimates of the historical pattern effect (Sect. 2.4) are excessive and that most current ECS estimates, including those based on process understanding and on emergent constraints as well as on historical warming – which (like those in S20 and L22) generally reflect the aforesaid weakening of climate feedback in GCMs – may be significantly biased upwards."

3. *I would recommend removing the second sentence of Section 3 so as not to sound antagonistic.*

I have, as suggested, deleted the sentence concerned.

**Responses to community comment CC1** (NB comment CC2 is just a very minor, non-relevant correction to CC1)

I thank Steven Sherwood for his helpful comments (hereafter S25) on my submitted manuscript (L25). My responses to points in it requiring input from the author (which I have numbered for ease of reference) are as follows.

1. *The current submission by Lewis on my short opinion piece with Chris Forest (SF24) doesn't contain much new information beyond Lewis's 2022 (L22) original critique of the 2020 WCRP assessment (S20).*

L25 is a comment on SF24, and the submitted manuscript therefore primarily addressed erroneous and contentious assertions in SF24, not in S20.

2. *Would it be possible for Lewis to clarify the impact on measure(s) of the posterior pdf (i.e. median/range) of each methodological difference or line of evidence between L22 and S20, to make this a more useful contribution?*

I accept that this suggestion would make L25 a more useful contribution , and in response to it substantial further information has now been incorporated in a new section 3 of L25: "Critique of SF24's Claims Regarding L22", which mentions each of the changes made in L22 and clarifies their impact, by methodological difference and line of evidence.

Some related changes have also been made to section 2, and a new sentence has been added in the Abstract in relation to the extra information now included in L25: " It also appraises L22's revisions to S20's methods and input assumptions, and considers how these have contributed to the lowering and narrowing of the ECS range."

Two additional sentences relating to adding section 3 have also been added in section 1:

"It identifies L22's contributions – through revisions to S20's input assumptions for each line of evidence, and other changes – to the lowering and narrowing of S20's ECS range."; and

"For clarity, it should be noted that while SF24 refers to ECS, S20 and L22 actually both estimate $S$, a proxy for ECS. In GCMs, $S$ is almost always slightly lower than ECS. The two terms are only distinguished herein when discussing their relationship."

3. *As it stands, the 2022 paper and this contribution may draw readers to infer that L22's ECS range differed from that of S20 due to corrected errors, ...*

I can't see that either L22 or the submitted L25 manuscript suggest that L22's ECS range differed from that of S20 due to corrected errors? The impact of correcting S20's likelihood estimation and of the change in prior were shown in Table 4 of L20 for all cases where S20 provides posterior pdfs. Reference to that table is now made in the revised L25 manuscript. Additionally, the revisions to S25 (in particular new section 3.1 and Table 1) make clear that correcting errors found in S20's likelihood estimates did not have a major effect on the ECS range. Also, the following sentences have been added in the first paragraph of section 2.1, mentioning actual errors in S20 analysis of PETM evidence that were corrected in L22:

"S20's likelihood estimates based on Paleocene-Eocene Thermal Maximum (PETM) evidence were similarly affected, although PETM evidence was not used in the main S20 results. Moreover, L22 pointed out that S20 used an uncertainty estimate that was a factor of ten lower than stated for PETM $CO_2$ forcing, due to a coding error."

*4. ... when in fact the reduction seems to be due to L22's one-sided re-interpretation of evidence and its narrowing due mainly to the use of a prior that is not well accepted by the broader community.*

The assertion that L22's narrowing of the ECS range was due mainly to the use of a prior that is not well accepted by the broader community is actually the opposite of the true effect of the computed Jeffrey's prior used in L22. As shown in L22 Tables 4 and 5, after correcting S20 likelihood estimation errors the 5–95% posterior range for S (S20's ECS proxy) was 2.25–4.85 K using S20's prior, but it was wider (and higher) at 2.3–5.05 using L22's prior. This point is covered in new section 3.1 and is now reiterated in section 4 (previously section 3).

S25 asserts that the reduction in the ECS range in L22 is due to one-sided re-interpretation of evidence. However, both the change in prior and the correction of S20's likelihood estimate errors increased the S20 median ECS estimate and 95% uncertainty bound, which is not suggestive of a one-sided approach to ECS estimation. In any event what is relevant is whether the revisions made in L22 produced improved estimates of the variables involved, not whether they were mostly in a direction that lowered estimated ECS.

As L22 Table 7 shows, just substituting, for S20's data values, the appropriate regression based estimate of $F_{2\times CO2}$ (as discussed in L25), and an ECS to S ratio derived by comparing estimates of those two values in each of 16 long $CO_2$ doubling or $CO_2$ quadrupling simulations, as set out in L22's Supplementary Information Section 5.3.1, rather than using S20's ECS to *S* ratio (which results in inconsistent estimation of S between different lines of evidence), reduces the 5–95% posterior range for *S* from 2.3–5.05 K to 1.95–4.35 K (using L22's prior and after correcting S20's likelihood errors in both cases). These changes, which accounted for approximately 40% of L22's total reduction in the median ECS estimate, involve methodological issues rather than reinterpretation of evidence. These changes are appropriate because they are in the first case mathematically correct, and in both cases are required to avoid inconsistent ECS estimation between different lines of evidence. These points are covered in new section 3.2. In addition, the discussion of $F_{2\times CO2}$ estimation methods in section 2.1 has been considerably reworded to make it easier to comprehend.

As regards the revisions made in L22 that did involve re-interpretation of evidence, in most cases they reflected newer evidence, rather than re-interpretation of evidence used by S20. In a few cases they were instead considered to be a fairer interpretation of existing evidence than S20's interpretation, with detailed justifications for the revised estimates being given in the L22 Supplementary Information.

All the various revisions made in L22 are now discussed and their effects on ECS estimation quantified by line of evidence in the new section 3, and summarised in Table 1, so the reader can form their own opinion on them. Reference is also made to tables in L22 that quantified the impact on measures of the posterior pdf of different data assumptions between L22 and S20 for each separate line of evidence, and for combined evidence ECS estimation by key classes of data differences. The L22 revisions that SF24 or S25 explicitly or implicitly challenge, suggesting that they were not in line with other scientists' appraisals, are those to the

historical pattern effect, historical aerosol forcing, and cloud feedback. These three revisions and their effects are discussed in sections 2.3, 2.4, 3.6, 3.7 and 3.8 of the revised manuscript.

5. ***Likelihood calculation****. It is claimed (following L22) that S20's likelihoods were "invalid." The argument is associated with the sampling method, which indeed was probably not optimal in S20. The sampling method matters by implying priors on uncertain variables such as radiative forcings.*

The sampling method S20 used to determine its likelihoods was undoubtedly invalid. S20 specified "priors on uncertain variables such as radiative forcings", which it used to derive its likelihoods. S20 section 2.3 stated: "For each of the independent variables except the $\lambda_i$, the prior PDF is specified by expert judgment using the available evidence about that quantity, without considering any other lines of evidence. These expert priors are given in the appropriate sections and are crucial in determining the historical and paleo likelihoods." A valid likelihood estimation method must use the already specified priors on relevant uncertain data variables, not imply different priors for them. The three likelihood estimation methods employed in L20 used the already specified priors on uncertain data variables, and produced likelihood estimates that were essentially identical to each other, but differed from the S20 estimates. Moreover, L22's Supplementary Information section S2 showed that if S20's arbitrary choice of $\Delta T$ as the unsampled independent variable was changed to another, a priori equally suitable independent variable, the estimated likelihood using S20's sampling method changed radically.

6. *In any case, as shown in Fig. 2d of L22, the discrepancy between the L22 and S20 sampling approaches had a negligible impact on the pdf of ECS, and so this issue didn't contribute to changing the results and hence isn't really relevant to SF24, which discusses advances post-2020.*

As discussed under 3. above, neither L22 nor L25 claimed that S20's likelihood estimate errors had a significant impact on its main ECS estimate. However, the error is indeed relevant to SF24, as its existence disproves SF24's claim concerning L22 that 'While this author claims "errors" in S20, looking carefully it appears these are differences in opinion on methodological choices and priors rather than errors'.

7. ***Priors****. Lewis argues in favour of "objective" priors as used by L22, which were not favoured by S20 nor by most Bayesian statisticians, for good reasons that aren't worth going into here. The main impact of L22's objective prior is to narrow the pdf—i.e., claim ECS to be known more confidently. The basis for this would be accepted by very few in the science community. Apart from that it is not new (Lewis would have used the same prior in 2020), and hence is also irrelevant to SF24 piece.*

As stated under 4. above, S25's claim that " The main impact of L22's objective prior is to narrow the pdf" is directly contradicted by results set out in L22, which show that the adoption of L22's objective prior for ECS actually widens and slightly shifts to higher values the PDF derived using S20's prior, as can be seen by comparing the 'All combined' estimates between Tables 4 and 5 of L22. The discussion of priors in section 4 of L25 is highly relevant to S24, which had a quite lengthy discussion of priors and argued for the use of subjective rather than "objective" priors on ECS or related variables.

Bayesian statisticians may perhaps disfavour noninformative "objective" priors because they don't always properly understand them, but there are few if any sound reasons for not using them where it is practicable to do so. Admittedly they can, however, be difficult to derive.

The relevant statements in L25 were correct, but as stated above the effect of adopting L22's "objective" prior is covered in new section 3.1 and is now reiterated in section 4.

8. ***Evidence****. It is changes to evidence or its interpretation (via best estimates or uncertainties on associated parameters) that must have shifted L22's ECS estimates downward. There were quite a few of these changes and it isn't clear which ones mattered most.*

*For parameters where AR6 estimates differed from S20 in a way that would push ECS lower, L22 chose the AR6 value (updating the GMST to GMAT relation; most forcing values including effective CO2 forcing; low cloud feedback). These seem to have small effects.*

*For parameters where AR6 values would push ECS higher, L22 did not use them, instead arguing in particular that the aerosol forcing and historical pattern effect were each smaller and better known than in either S20 or AR6. Given the range of recent published studies, this doesn't appear to be consistent with the views of researchers working on this topic or their evidence. S20 had allowed a large uncertainty on the pattern effect including the possibility of both small and large values, which continue to be supported by new studies (although Modak and Mauritsen 2023 find a slightly smaller value, Myers et al. 2023, Alessi and Rugenstein 2023, Olonscheck and Rugenstein 2023, and Liang et al. 2023 continue to find substantial pattern-induced changes in the planetary energy budget that imply a larger future warming than would be inferred from historical fits of simple energy-balance models).*

These points are dealt with in sections 2.3, 2.8 and 3.7 of the revised manuscript and/or in the foregoing response to item 4. Some additional responses on specific points are also relevant.

Substituting updated estimates from IPCC AR6 of the ratio of global near surface air temperature to estimates thereof with warming over the ocean based on sea surface temperature, and of changes in historical forcings other than for aerosols involved substitution of newer, IPCC derived estimates, rather than reinterpretation of existing evidence. Incidentally, although SF25 states that L22 adopted the AR6 value for low cloud feedback, as stated in L22 it was actually adopted from Myers et al. (2021), but with their uncertainty standard deviation increased to match that in S20.

S25's claim that L22 argued that the historical pattern effect was better known than in S20 is incorrect; L22 used the same standard deviation for its estimate of 0.35 $Wm^{-2}K^{-1}$ as S20 did for their slightly higher estimate of 0.5 $Wm^{-2}K^{-1}$. Moreover, as now made clear in L25 section 3.7 and Table 1, the revision in L22 to S20's historical pattern effect estimate had almost no effect on its main combined-evidence climate sensitivity range. The discussion of historical pattern effect estimates in L25 has now been revised by inclusion of a recent study (Modak and Mauritsen 2023) that supports L22's revision to S20's estimate. S25 cites four recent studies that find substantial pattern effect induced changes in the planetary energy budget (Myers et al. 2023, Alessi and Rugenstein 2023, Olonscheck and Rugenstein 2023, and Liang et al. 2023). However, none of these provide evidence against L22's revision of S20's estimate. Neither Alessi and Rugenstein (2023) nor Liang et al. (2024) actually give an estimate of the pattern effect magnitude, and (like two of the three pattern effect studies cited in SF24) the other two studies only give estimates based on a relatively short period (respectively 1980–2014 and 2001–2021), not over the full historical period.

S25 also questions the revision made in L22 to S20's historical aerosol forcing, although neither S25 or SF24 cite any specific new estimates of total historical aerosol forcing.

In the revised manuscript, the text in new section 3.8 states that both historical aerosol forcing and cloud feedback remain poorly constrained and that L22's revisions to the S20 aerosol forcing and cloud feedback estimates should be regarded as subject to significant uncertainty. However, even if in the light of later evidence L22's revision to S20's aerosol forcing estimate turned out to be unsupportable and so did the L22 revision to S20's historical pattern effect estimate (despite no good evidence to date suggesting that the latter was unjustified: see section 2.4), the effect on L22's ECS range would very small, as stated in new section 3.7 – just an increase of 0.1 K in the 95% uncertainty bound and a change of 0.03 K in the median estimate.

9. *Forcing. L22 also multiplied the CO2 forcing by 0.85, which doesn't come from any study, claiming that the regression intercept should be used rather than the equilibrium change used by S20. But this introduces an inconsistency between forcing and feedback (even though Lewis claims the opposite), since S20 effectively inferred both feedbacks and forcings from equilibrium response estimates, not Gregory-type regression slopes.*

S25 disputes L22's multiplication of the best estimate of [doubled] $CO_2$ forcing by 0.85; the multiplier used was in fact 0.86. This came from calculations using a large ensemble of climate models. The rational was explained in section S1 of L22's Supplementary Information and the calculated parameters for each climate model were set out in its Table S1.

S25 claims that S20 effectively inferred both forcing and feedback from equilibrium response estimates, not from Gregory-type regression slopes. That is self-evidently not correct in relation to S20's process and historical estimates of feedback and climate sensitivity ($S$).

S20's definition of $S$ directly corresponds to climate sensitivity estimated from a Gregory regression. It says (S20 section 2.1):

"$S$ [is] derived from system behavior during the first 150 years following a (hypothetical) sudden quadrupling of CO2. During this time the system is not in equilibrium, but regression of global mean top-of-atmosphere (TOA) energy imbalance onto global mean near-surface air temperature (SAT), extrapolated to 0 imbalance, yields an estimate of the long-term warming valid if the average feedbacks active during the first 150 years persisted to equilibrium (Gregory et al., 2004)."

Moreover, S20 states (section 3.1.3) that its target process feedback estimates were values for feedbacks acting over the 150 years following an increase in $CO_2$ – that is, derived on a basis consistent with feedbacks in 150 year abrupt4xCO2 GCM simulations. And the historical pattern effect estimate, used to adjust feedback estimated from historical data to that needed to estimate $S$, represents the difference in GCMs between estimates of feedbacks that operated over the historical period and feedbacks operating over150 year abrupt4xCO2 GCM simulations. Since the process and historical evidence based feedback estimates thus did correspond to Gregory-type regression slopes, the estimates of $S$ derived from them should also have used forcing estimates corresponding to Gregory-type regression slopes.

Since S20's initial paleoclimate evidence based feedback estimates were indeed equilibrium response based (before adjustment from an ECS equilibrium basis to a 150-year regression basis estimate of $S$), L22 did not apply its 0.86 multiplier to $CO_2$ ERF in paleoclimate cases.

10. *A simple consistency "sanity check" can be obtained by comparing the estimated parameters and resulting ECS pdf with those of CMIP GCMs. S20's best-estimated overall feedback parameter was only slightly stronger than the median from CMIP GCMs (S20 Fig. 4) implying a slightly lower ECS, while the forcing was slightly stronger hence slightly raising ECS; together these imply a median ECS from process evidence that should have closely matched that from CMIP GCMs. It indeed did, with both coming in at about 3.1K.*

S25 claims that S20's best-estimated overall feedback parameter ($1.3$ $Wm^{-2}K^{-1}$) was only slightly stronger than the median from CMIP GCMs (S20 Fig. 4) implying a slightly lower ECS. In fact, taking the average of the CMIP5 GCM feedback estimates in S20 Fig.4, other than the Vial et al (2013) estimate of $-1.7$ $Wm^{-2}K^{-1}$, gives a CMIP5 feedback estimate of $1.03$ $Wm^{-2}K^{-1}$ – close to the AR5 Table 9.5 estimate of $1.1$ $Wm^{-2}K^{-1}$. (I exclude the old Vial et al estimate as it is clearly unreliable: they *inter alia* estimated feedback in the IPSL-CM5A-LR model to be nearly double that in the INMCM4 model, despite the former model having an estimated ECS nearly double that of INMCM4 [IPCC AR5 Table 9.5]) Averaging that $1.03$ $Wm^{-2}K^{-1}$ value and the S20 Fig.4 estimate for CMIP6 GCMs of $1.09$ $Wm^{-2}K^{-1}$ gives an average estimate of $1.06$ $Wm^{-2}K^{-1}$, marginally higher than the $0.98$ $Wm^{-2}K^{-1}$ median feedback for the CMIP5 and CMIP6 combined ensemble used to derive L22's 0.86 multiplier. The median ECS estimates for CMIP5 and CMIP6 GCMs with complete entries in AR6 Table 7.SM.5 are respectively 2.83 and 3.72 °C. Multiplying the average of those ECS estimates by the ratio of the $1.06$ $Wm^{-2}$ $K^{-1}$ average CMIP5 and CMIP6 feedback estimates derived from S20 Fig.4, to S20's $1.30$ $Wm^{-2}$ $K^{-1}$ feedback estimate, implies a S20-feedback-derived ECS estimate of 2.67 K. By comparison, dividing S20's $4.0$ $Wm^{-2}$ $CO_2$ ERF value by its$1.30$ $Wm^{-2}$ $K^{-1}$ feedback estimate gives a higher ECS estimate of 3.08 K, but after applying L22's 0.86 adjustment factor this reduces to 2.65 K – almost identical to 2.67 K.

The above "consistency check" calculation therefore provides further support for the application of L22's 0.86 multiplier to the estimated actual doubled $CO_2$ forcing, when estimating ECS using feedback estimates that are derived, as in S20, on a basis consistent with their values over 150 year abrupt $CO_2$ quadrupling simulations. Moreover, applying such a multiplier is undoubtedly theoretically correct.

The discussion in L25 (and L22) of the rationale for the multiplier is accordingly correct, however its wording was sub-optimal and has now been revised (in section 2.1) to make it easier to follow.

11. *As to historical evidence, Lewis's new contribution seems to confuse the question of whether it was important in S20 vs. whether its influence has changed. While S20 found that the historical evidence did almost nothing to discount high ECS values, the central estimate based on historical evidence alone dropped from 5.82K in S20 to 2.16K in L22 (according to L22), which suggests historical evidence did become a lot more constraining for L22. This, combined with Lewis' claim that aerosol forcing change had a small effect, led to SF24's deduction that the pattern effect revision must have been important. However SP24 might have got this wrong—which is why it would be valuable for Lewis to clarify what is really going on in L22. The revision of the historical likelihood implies that historical evidence did become an additional constraint against high ECS in L22.*

S25 asserts that the revision of S20's historical likelihood in L22, with a resulting large drop compared to S20 in the central ECS estimate from historical evidence alone, implies that historical evidence did become a significant additional constraint against high ECS in L22. However, as L22 Figure 5 shows, at L22's combined evidence 95% bound of 3.2 K its historical likelihood had not declined very far from its peak value, and even at higher climate sensitivity values the rate of its decline was very much slower than for the likelihood from either process or paleoclimate evidence. That is why historical evidence was a weaker constraint on high ECS than either process or paleoclimate evidence (L22 Tables 6 and 8). Although it did provide a weak constraint – omitting historical evidence entirely increased L22's 95% ECS bound by 0.2 K, from 3.2 K to 3.4 K – only 0.1 K of this reduction was due to the historical aerosol forcing and pattern effect magnitudes being revised down in L22. These points are discussed in new section 3.7.

12. *Changes in process evidence also look to play some role in the decline of ECS in L22. L22 reduced the low-cloud feedback compared to S20 (a small reduction also noted by SF24) but failed to increase the feedback due to high clouds (no longer thought to be negative, see e.g. McKim et al. 2024 Nat. Geosci, Sokol et al., 2024, Nat. Geosci. and Raghuraman et al. 2024 JGR and earlier studies) or cloud phase (a negative feedback that has is likely overestimated in GCMs due to incorrect supercooled water, although this remains uncertain; see e.g. Cesana et al. 2024, Zhao et al. 2024).*

S25 is correct in asserting that changes in process evidence values in L22, very largely in estimated low-cloud cloud feedback, play a role in the decline in ECS in L22. The revised manuscript now refers (section 3.8) to two of the three cloud feedback studies cited in SF24; the third study appears to estimate aerosol forcing, not cloud feedback. S25 cites further evidence from studies that suggest non-low cloud feedback is higher than assumed in both S20 and L22. The studies cited were all published in 2024, so could not have been taken account of in L22. Cloud feedback remains poorly constrained and other studies may reach different conclusions. The statement in the discussion in new section 3.8 that L22's revision to the cloud feedback estimate should be regarded as subject to significant uncertainty is considered adequate to also cover the findings of these 2024 studies, particularly as the effect of excluding changes in cloud feedback on L22's combined evidence ECS estimate is modest – and remains modest even if the L22 historical aerosol forcing and/or pattern effect estimates are also reverted back to their S20 values. These effects are now quantified in new sections 3.6 and 3.8. If the L22 cloud feedback, historical aerosol forcing and pattern effect estimates are all reverted back to their S20 values, the ECS range would become 1.65–3.6 K (median 2.34 K). So only a small proportion of the total reduction from S20's range of 2.3–5.05 K (median 3.23 K), after correcting its likelihoods and adopting L22's prior, to L22's posterior range of 1.55–3.2 K (median 2.16 K) would be called into question. The remaining reduction represents only 13% and 22% respectively

of the total reduction in L22 of the ECS range lower and upper bounds, and only 17% of the reduction in the median.

13. *The Zhao et al. study also implies stronger indirect cooling by aerosols than previously expected, in opposition to the aerosol forcing revision of L22, and few new constraints are available for mixed-phase or ice clouds which implies that a large uncertainty remains in the overall forcing, potentially larger than even was used in S20 since the Bellouin et al. / WCRP assessment that provided it did not fully account for uncertainties from deep convective cloud effects.*

This comment concerns a recent study suggesting that historical aerosol forcing could be even more uncertain than assumed in S20. However, this is not based on observations, but rather on simulations by two GCMs. The average ECS of the two GCMs is over 4.5 K – which S20 says (in its Abstract) there is strong evidence against.

Moreover, as explained in section 2.3 of the revised manuscript, even using S20's assumption regarding aerosol forcing there is a substantial probability that the denominator of the energy budget formula for estimating ECS is small or negative. That results in the historical likelihood only declining fairly modestly at high ECS values (see L22 Fig. 5(c)), thus providing very little constraint on high ECS values. Adding even greater uncertainty at the stronger aerosol forcing end of the distribution would therefore have very little effect on the upper bound of L22's ECS estimate. Nor would adding greater uncertainty at the weaker aerosol forcing end of the distribution have more than a minor effect on reducing the lower bound of L22's ECS estimate, as the process and paleoclimate evidence likelihoods fall very sharply at ECS levels below L22's 5% uncertainty bound.

In view of the effect on L22's combined evidence ECS estimate of even more uncertain aerosol forcing than that per S20's estimate almost certainly being very minor, no changes to the L25 manuscript have been made in this connection.

**References (other than for citations repeated here because they were cited in S25)**

Annan JD, Hargreaves JC, Mauritsen T. A new global surface temperature reconstruction for the Last Glacial Maximum. Climate of the Past. 2022 Aug 18;18(8):1883-96.

Modak, A., and Mauritsen, T.: Better constrained climate sensitivity when accounting for dataset dependency on pattern effect estimates, Atmospheric Chemistry and Physics, 23(13), 7535-7549., 2023, https://doi.org/10.5194/acp-23-7535-2023.

Myers T.A., Scott R.C., Zelinka M.D., Klein S.A., Norris J.R., Caldwell P.M.: Observational constraints on low cloud feedback reduce uncertainty of climate sensitivity, Nat Clim Chang 11(6):501–507, https://doi.org/10.1038/s41558-021-01039-0, 2021.

Osman, M.B., Tierney, J.E., Zhu, J., Tardif, R., Hakim, G.J., King, J. and Poulsen, C.J., 2021. Globally resolved surface temperatures since the Last Glacial Maximum. *Nature*, *599*(7884), pp.239-244. https://www.nature.com/articles/s41586-021-03984-4

Tierney, J.E., Zhu, J., King, J., Malevich, S.B., Hakim, G.J. and Poulsen, C.J.: Glacial cooling and climate sensitivity revisited, Nature, 584(7822), pp.569-573, 2020, https://doi.org/10.31223/osf.io/me5uj.

Vial, J., Dufresne, J.-L., & Bony, S.: On the interpretation of inter-model spread in CMIP5 climate sensitivity estimates, Climate Dynamics, 41, 3339–3362, 2013, https://doi.org/10.1007/s00382-013-1725-9